# Consistent Data Distribution Sampling for Large-scale Retrieval

## Abstract

Retrieving candidate items with low latency and computational cost is important for large-scale advertising systems. Negative sampling is a general approach to model million-scale items with rich features in the retrieval. The training-inference inconsistency of data distribution brought from sampling negatives is a key challenge. In this work, we propose a novel negative sampling strategy Consistent Data Distribution Sampling (CDDS) to solve such an issue. Specifically, we employ a relative large-scale of uniform training negatives and batch negatives to adequately train long-tail and hot items respectively, and employ high divergence negatives to improve the learning convergence. To make the above training samples approximate the serving item data distribution, we introduce an auxiliary loss based on an asynchronous item embedding matrix over the entire item pool. Offline experiments on real datasets achieve SOTA performance. Online experiments with multiple advertising scenarios show that our method has achieved significant increases in GMV. The source code will be released in the future.

## 1 Introduction

Industrial search systems, recommendation systems, and advertising systems generally contain a large scale of users and items. To handle the millions and even billions of items, such systems usually comprise the matching stage and the ranking stage. The matching stage, which is also called retrieval, aims to retrieve a small subset of item candidates from the large item pool. Based on the thousand-scale retrieval subset, the ranking stage concentrates on the specific ranks of items for the final display. Since the retrieval task should consider both accuracy and efficiency, the two-tower architecture is the main-stream matching method widely used in most industrial large-scale systems(Yi et al., 2019).

In a two-tower retrieval mechanism, a million-scale to billion-scale item embedding can be prepared in advance and updated in a regular cycle when serving. FAISS (Johnson et al., 2019) is usually employed to quantize the vectors. With the efficient nearest neighbors search mechanism, retrieving top similar items from the entire item pool for a given query can be low latency. But the same procession is extremely expensive in the training phase. Especially entire item embeddings change a lot along with the training steps and are hard to feed-forward in every training step. On this condition, the inconsistency between the training data distribution and inference data distribution can not be ignored. What's more, the inconsistency is much more serious when the scale of the items grows.

Since the training-inference inconsistency has a great impact in practice, recent research explored various ways to construct negative training instances to alleviate the problem. Faghri et al. (2017) employ contrastive learning to select hard negatives in current or recent mini-batches. Xiong et al. (2020) construct global negatives using the being-optimized dense retrieval model to retrieve from the entire corpus. Grbovic & Cheng (2018) encode user preference signals and treats users' rejections as explicit negatives. Yang et al. (2020) propose mixed negative sampling to use a mixture of batch and uniformly sampled negatives to tackle the selection bias.

Totally, uniform sampling negatives are often easy to distinguish and yield diminishing gradient norms (Xiong et al., 2020). Batch negatives lead to an insufficient capability when retrieving long-tail items and over-suppress those hot items which are interacted frequently. The current mixed sampling method does not clarify the role and sampling distribution between easy negative samples

and hot negative samples. The combination of various sampling methods is usually well-designed for the respective case rather than a general situation.

Although previous work sought to empirically alleviate the training-inference inconsistency problem in the matching stage, directly approximating the matching probability over the entire item data distribution is absent. Complex sampling methods make it difficult to analyze the effect of a single negative sample on the selection bias. Empirical approaches which have achieved improvement in efficiency metrics are uncertain about whether the selection bias is actually reduced due to the difficulty of analysis. Since only the non-sampling method is truly free of selection bias, optimizing the selection bias becomes possible if the retrieval probability of a single item can be approximated over the entire item pool.

To this end, we propose a novel negative sampling approach called Consistent Data Distribution Sampling, CDDS[1] for brevity. To adequately train long-tail and hot items respectively, we employ a relative large-scale of uniform training negatives and batch negatives. To improve the learning convergence, we employ global approximate nearest neighbor negatives as high divergence negatives. By maintaining an asynchronous item embedding matrix, we calculate the loss not only between the query embedding and the sampled items but also over the entire item pool. Directly approximating the matching probability over the entire item data distribution, our theoretical analysis and experiments show that CDDS achieves a consistent training-inference data distribution, a fast learning convergence, and the state-of-the-art performance.

In summary, the main contributions of this work are as follows:

- We analyze the bottlenecks of different negative sampling strategies and show that introducing those unsampled vast majority items can alleviate the training-inference inconsistency and resolve the bottlenecks.

- We propose CDDS which adequately trains long-tail items, improves the learning convergence, and directly approximates the matching probability over the entire item data distribution.

- Our theoretical analysis proves that CDDS achieves a consistent training-inference data distribution. Extensive experiments on real-world datasets achieve the state-of-the-art results. Online experiments with multiple advertising scenarios show that our method has achieved significant increases in GMV and adjust cost.

## 2 RELATED WORK

### 2.1 TWO-TOWER MODELS

The two-tower architecture is popular in industrial retrieval, which learns query and item vector representations separately in two forward networks. This framework has been widely used in text retrieval (Huang et al., 2013; Yang et al., 2020; Hu et al., 2014), entity retrieval (Gillick et al., 2019), and large-scale recommendation (Cen et al., 2020; Covington et al., 2016; Li et al., 2019). Our work is orthogonal to existing complex user representation network architectures such as CNN (Hu et al., 2014; Shen et al., 2014) and multi-interest models (Li et al., 2019; Cen et al., 2020), which can also benefit from our optimal sampling strategy.

### 2.2 NEGATIVE SAMPLING METHODS FOR TWO-TOWER MODELS

Covington et al. (2016) and Li et al. (2019) formulate the retrieval task as an extreme multi-class classification with a sampled softmax loss. Although the challenge of efficient training and the training-inference inconsistency seems to be solved, these methods rely on a predetermined item pool and are not applicable when sampling with complex distributions (Yang et al., 2020).

Chen et al. (2020) show that an enlarged negative pool significantly benefits sampling. In text retrieval, ANCE (Xiong et al., 2020) constructs global negatives from the entire corpus by using an asynchronously updated ANN index, theoretically showing that uninformative items yield dimin-

---

[1] we will release the code once the paper is accepted.

ishing gradient norms. That is to say, diminishing gradient norms matter in introducing an enlarged negative pool.

MNS (Yang et al., 2020) finds it important to reduce the selection bias brought by batch negative sampling and reduces such bias by additionally sampling uniform negatives from the item pool. All items in the item pool have a chance to serve as training negatives and the sampling distribution is more flexible. Although the performance of fresh and long-tail items is better, a small scale of sampling uniform negatives is insufficient with a huge-scale item pool and a large scale of sampling uniform negatives easily leads to high latency and computational cost. Meanwhile, batch negatives are not a complete substitute for hard negatives to achieve a fast learning convergence.

## 3 PROBLEM FORMULATION

We consider a recommender/search/advertising system with a query set $\mathbb{U} = \{u_1, u_2, \ldots, u_M\}$ and an item set $\mathbb{V} = \{v_1, v_2, \ldots, v_N\}$, where $M$ is the number of users and $N$ is the number of items. Given a query $u$, its positive feedback on items forms the training positive set $\mathbb{V}^+$. For the given query, the matching stage aims at efficiently retrieving a possible subset $\hat{\mathbb{V}}^+$ from the universe set $\mathbb{V}$, where the desired candidate items number $|\hat{\mathbb{V}}^+| \ll |\mathbb{V}|$.

In industrial-scale applications, the number of items can be enormously large and the training data is likely to be very sparse for most queries and items. To tackle the above challenges, the two-tower model with towers referring to encoders based on DNN is usually applied (Yu et al., 2021). The two-tower model learns latent representations of the given query and item separately. We denote the two towers by functions $f(x_u; \boldsymbol{\theta})$ and $g(x_v; \boldsymbol{\theta})$, where $f()$ is the query tower model, $g()$ is the item tower model, $x_u$ and $x_v$ map query and item features, and $\boldsymbol{\theta}$ is the whole parameters. Then the matching score of the corresponding item becomes:

$$\varepsilon(u, v) = sim(\boldsymbol{u}, \boldsymbol{v}), \tag{1}$$

where $\boldsymbol{u} = f(x_u; \boldsymbol{\theta})$, $\boldsymbol{v} = g(x_v; \boldsymbol{\theta})$, and the similarity function ($sim()$) is often simply cosine or dot product. Then we can calculate the matching score of the item $v_i$ in softmax probability:

$$P(v_i|u) = \frac{e^{\varepsilon(u,v)}}{\sum_{v_i \in \mathbb{V}} e^{\varepsilon(u,v_i)}}, \tag{2}$$

and the items with top values will be the candidates.

## 4 METHODOLOGY

In this section, we discuss what difficulties are encountered in calculating the matching score $P(v|u)$ and provide theoretical analyses of how CDDS approximates the $P(v|u)$ with the sampled negatives and the universe set $\mathbb{V}$.

### 4.1 TRAINING-INFERENCE INCONSISTENCY

Constructing the positive feedback items $\mathbb{V}_r^+$ as training positives and the other exposed items $\mathbb{V}_r^-$ as training negatives is a common approach in the ranking stage. With a well-working ranking model, the items exposed are often with top ranking scores and is not a proper matching training negatives consider that:

$$P(v|u, v \in \mathbb{V}_r^+) > P(v|u, v \in \mathbb{V}_r^-) > P(v|u, v \in \mathbb{V}\backslash\mathbb{V}_r^+\backslash\mathbb{V}_r^-). \tag{3}$$

$\mathbb{V}_r^+$ and $\mathbb{V}_r^-$ are all come from the matching stage and are the top part of the matching candidates. Training a retrieval model like a ranking model becomes increasingly unacceptable as the $|\mathbb{V}\backslash\mathbb{V}_r^+\backslash\mathbb{V}_r^-|$ grows. Due to the ambiguity of user feedback, an item without positive feedback is not an exactly undesired candidate. Sometimes, to serve the ranking model better, it is useful to define the ranking tops as training positives $\mathbb{V}_r^+ \cup \mathbb{V}_r^-$ in the matching stage. Since the number of positive items is not much bigger, it is feasible to pick positive samples that are close to the specific application. In this condition, we define the well-picked training positives as $\mathbb{V}^+$ that are better than all the others in the item pool. As a result of retrieving a possible subset $\hat{\mathbb{V}}^+$ from the universe set $\mathbb{V}$ in the

serving phase, the training negative set corresponding to the training positive set is $\mathbb{V}^- = \mathbb{V} \backslash \mathbb{V}^+$. To describe the training positives accurately, we must focus on the long-tail majority $\mathbb{V}^-$.

When the number $|\mathbb{V}^-|$ is not extremely enormous, training a retrieval model as a multi-classification task with the whole $\mathbb{V}^-$ is an alternative approach. But as the number of the training negatives grows, training with $\mathbb{V}^-$ becomes a high computational cost. In the inference, all items $\mathbb{V}$ can be prepared in advance and updated in a regular cycle even if the number is millions to billions. Thanks to the efficient library for similarity search in billions of vectors, FAISS (Johnson et al., 2019) for example, embedding search with all items in $\mathbb{V}$ is fast and accurate. Since the same procession is extremely expensive in the training phase, it is impossible to train with all negatives. Sampling a negative subset $\hat{\mathbb{V}}^-$ becomes a natural choice and it leads to the training-inference inconsistency on the data distribution. The training-inference inconsistency can be represented as

$$\begin{cases} \hat{\mathbb{V}}^- \subset \mathbb{V}^- \\ |\hat{\mathbb{V}}^-| \ll |\mathbb{V}^-| \end{cases}. \tag{4}$$

### 4.2 IMPACT OF INCONSISTENT DATA DISTRIBUTION

Given the query $u$, the best parameters $\boldsymbol{\theta}^*$ with the ideal training negatives $\mathbb{V}^-$ is:

$$\boldsymbol{\theta}^* = \operatorname{argmin}_{\boldsymbol{\theta}} \sum_u \sum_{v^+ \in \mathbb{V}^+} \sum_{v^- \in \mathbb{V}^-} l(\varepsilon(u, v^+), \varepsilon(u, v^-)). \tag{5}$$

where $l()$ denotes the loss function. After adopting negative sampling, the optimization goal becomes:

$$\hat{\boldsymbol{\theta}}^* = \operatorname{argmin}_{\boldsymbol{\theta}} \sum_u \sum_{v^+ \in \mathbb{V}^+} \sum_{v^- \in \hat{\mathbb{V}}^-} l(\varepsilon(u, v^+), \varepsilon(u, v^-)). \tag{6}$$

Without sampling, for every positive item $v^+$ trained, the negative item $v^-$ is also trained once. In the uniform sampling case, for every time the $v^+$ is trained, $v^-$ is trained with an expectation of $\frac{|\hat{\mathbb{V}}^-|}{|\mathbb{V}^-|}$ times. As a result of $|\hat{\mathbb{V}}^-| \ll |\mathbb{V}^-|$, most items are not trained enough. When calculating softmax loss function, we can apply a weight to adjust the loss of training samples. However, to ensure that most items are trained adequately, the time and computational cost are not acceptable.

In order to make the $\hat{\mathbb{V}}^-$ as close as the $\mathbb{V}^-$, it seems that we should increase the number of uniform samples as much as possible. But Xiong et al. (2020) show that a negative instance with a larger gradient norm is more likely to reduce the training loss. Let $l(v^+, v^-) = l(\varepsilon(u, v^+), \varepsilon(u, v^-))$, Xiong et al. (2020) and Katharopoulos & Fleuret (2018) proved that

$$l(v^+, v^-) \to 0 \Rightarrow \|\nabla_{\phi_L} l(v^+, v^-)\|_2 \to 0 \Rightarrow \|\nabla_{\boldsymbol{\theta}_t} l(v^+, v^-)\|_2 \to 0, \tag{7}$$

where $\|\nabla_{\phi_L} l(v^+, v^-)\|_2$ denotes the gradient with respect of the last layer and $\|\nabla_{\boldsymbol{\theta}_t} l(v^+, v^-)\|_2$ denotes the gradient at training step $t$. As the number of desired candidates is small, a large scale of sampling uniform negatives is easily to be uninformative and yields diminishing gradients.

### 4.3 NEGATIVE SAMPLING APPROACH

Based on the above analysis, we propose a Consistent Data Distribution Sampling approach that aims to alleviate the training-inference inconsistency and achieve a fast learning convergence.

In order to close to the serving item data distribution, we employ a relative large-scale of uniform negatives. The number of uniform negatives is determined by both the number of queries and the number of items to ensure each item is trained as much as possible when the training phase is complete. A larger number of training negatives means a higher computational cost. So we make uniform sampling during the training period and share the same uniform negatives in a mini-batch.

In order to achieve a fast learning convergence, hard negatives sampling is useful. A simple but effective approach is using the positive items in a query batch $\{u_1, u_2, \ldots, u_B\}$ as shared negatives for all queries in the same batch (if a shared negative happens to be the positive item of the given query, it is filtered for the given query). Yang et al. (2020) show that a mixture of batch and uniformly sampled negatives significantly improves retrieval quality. However, the batch negatives are mainly those items that are frequently interacted with and are popular rather than hard.

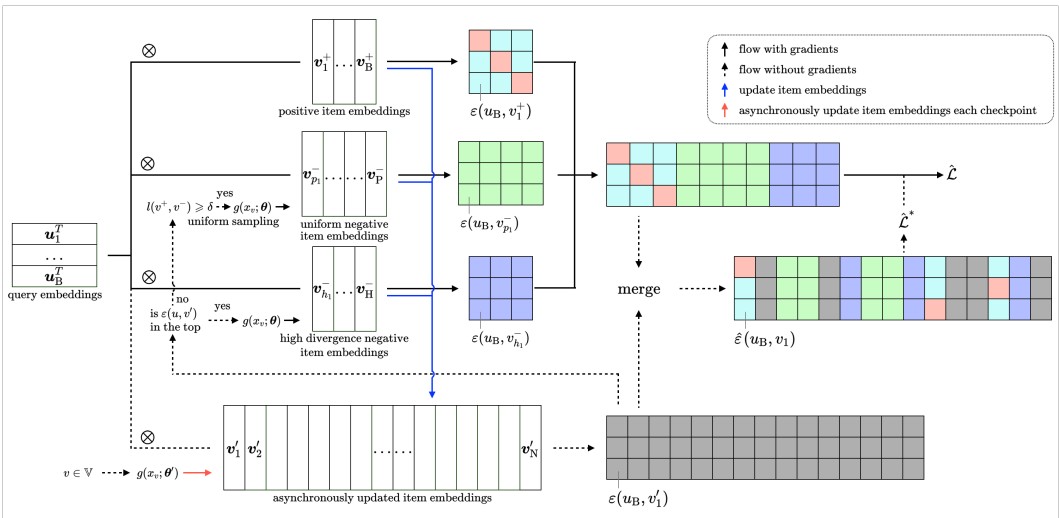

Figure 1: The workflow of the proposed CDDS method. We employ a relative large-scale of uniform training negatives, batch negatives, and high divergence negatives as mixture negatives. By updating item embeddings with higher gradients at higher frequencies, we approximate the similarity score over the entire item pool. With the auxiliary loss function $\hat{\mathcal{L}}^*$, we refine the loss function $\hat{\mathcal{L}}$ among the sampled items.

Similar to Xiong et al. (2020), to train truly hard and informative samples, we imply an asynchronously updated ANN index of the item pool and maintain an inferencer that parallelly computes the item embedding with a recent checkpoint. We introduce the top items for each query as high divergence negatives. The difference is that we de-duplicate these top items and use them as shared negatives within the batch. That is, we guarantee that each query is trained with high divergence items, but we do not strictly limit that it is only trained with hard negatives.

A mixture of uniform negatives $\mathbb{V}_p^- = \{v_{p_1}^-, v_{p_2}^-, \ldots, v_P^-\}$, batch negatives $\mathbb{V}_b^- = \{v_1^+, v_2^+, \ldots, v_B^+\}$ and high divergence negatives $\mathbb{V}_h^- = \{v_{h_1}^-, v_{h_2}^-, \ldots, v_H^-\}$ form the final negatives $\hat{\mathbb{V}}^-$. All negatives are shared in a batch. Let $y_i$ be the label of item $v_i$, and the loss function with inconsistency is as follows:

$$\mathcal{L} = \sum_{v_i \in \mathbb{V}^+ \cup \hat{\mathbb{V}}^-} -[y_i \log(\hat{y}_i) + (1 - y_i) \log(1 - \hat{y}_i)], \tag{8}$$

where

$$\hat{y}_i = \frac{e^{\varepsilon(u,v)}}{\sum_{v_i \in \mathbb{V}^+ \cup \hat{\mathbb{V}}^-} e^{\varepsilon(u,v_i)}}. \tag{9}$$

When we use a large-scale uniform negative sampling, our training negatives are close to the serving data and it is inefficient in gradient updating. When we introduce items with high training divergence negatives, training-inference inconsistency is inevitable. From the training-inference inconsistency perspective, we desire the loss function to be:

$$\mathcal{L}^* = \sum_{v_i \in \mathbb{V}} -[P(v_i|u) \log(P(v_i|u)) + (1 - y_i) \log(1 - P(v_i|u))]. \tag{10}$$

To approximate the matching score of the item in softmax probability $P(v_i|u)$, we need to find an approach to approximate the similarity score $\varepsilon(u, v)$ and the item embedding $g(x_v; \boldsymbol{\theta})$. On a coarse-grained approximation, we can decompose the similarity score $\varepsilon(u, v)$ into two components:

$$\hat{\varepsilon}(u, v) = \begin{cases} sim(f(x_u; \boldsymbol{\theta}), g(x_v; \boldsymbol{\theta})), & \text{if } v \in \mathbb{V}^+ \text{ or } v \in \hat{\mathbb{V}}^- \\ sim(f(x_u; \boldsymbol{\theta}), g(x_v; \boldsymbol{\theta}')), & \text{else} \end{cases}, \tag{11}$$

where $\boldsymbol{\theta}'$ is the parameters of the recent checkpoint. Along with the training, the difference between $\boldsymbol{\theta}$ and $\boldsymbol{\theta}'$ becomes larger ($g(x_v; \boldsymbol{\theta}) \not\sim g(x_v; \boldsymbol{\theta}')$), so we cannot directly replace the retrieval probability $\varepsilon(u, v)$ with $sim(f(x_u; \boldsymbol{\theta}), g(x_v; \boldsymbol{\theta}'))$.

Let's consider a SGD step with importance sampling (Alain et al., 2015):

$$\boldsymbol{\theta}_{t+1} = \boldsymbol{\theta}_t - \eta \frac{1}{|\hat{\mathbb{V}}^-|p_{v^-}} \nabla_{\boldsymbol{\theta}_t} l(v^+, v^-), \tag{12}$$

where $\boldsymbol{\theta}_t$ denotes the parameters at $t$-th step, $\boldsymbol{\theta}_{t+1}$ denotes the parameters one step after, $\eta$ denotes the learning rate, and $p_{v^-}$ denotes the sampling probability of negative instance. From equation 7 and equation 12, to make $\hat{\varepsilon}(u, v) \approx \varepsilon(u, v)$:

- We apply a threshold $\delta$ and ignore the uninformative negatives ($l(v^+, v^-) < \delta$) which yields diminishing gradient norms.

$$l(v^+, v^-) \to 0 \Rightarrow \boldsymbol{\theta}_{t+1} - \boldsymbol{\theta}_t \to 0 \Rightarrow \hat{\varepsilon}(u, v) \approx \varepsilon(u, v) \tag{13}$$

- We introduce the top items with high $sim(f(x_u; \boldsymbol{\theta}), g(x_v; \boldsymbol{\theta}'))$ as high divergence negatives every batch. Meanwhile, all high divergence negative item embeddings $[\boldsymbol{v}_{h_1}^-, \boldsymbol{v}_{h_2}^-, \ldots, \boldsymbol{v}_{\mathrm{H}}^-]$ are updated to the asynchronous item matrix $[\boldsymbol{v}_1', \boldsymbol{v}_2', \ldots, \boldsymbol{v}_N']$. So the item embeddings with a high variation $\boldsymbol{\theta}_{t+1} - \boldsymbol{\theta}_t$ are updated every batch.

- We make uniform sampling only from the remained subset. Embeddings with a relatively high variation $\boldsymbol{\theta}_{t+1} - \boldsymbol{\theta}_t$ are updated with a stable frequency.

With such a process, hot item embeddings, high divergence item embeddings, and relative large-scale uniform item embeddings are all updated along with the variance of the parameters. Further, the approximated retrieval probability is

$$\hat{P}(v_i|u) = \frac{e^{\hat{\varepsilon}(u,v)}}{\sum_{v_i \in \mathbb{V}} e^{\hat{\varepsilon}(u,v_i)}}, \tag{14}$$

and the desired loss function $\mathcal{L}^*$ is transformed into an auxiliary loss function

$$\hat{\mathcal{L}}^* = \sum_{v_i \in \mathbb{V}} -[\hat{P}(v_i|u) \log(\hat{P}(v_i|u)) + (1 - y_i) \log(1 - \hat{P}(v_i|u))]. \tag{15}$$

Considering that we cannot apply the backpropagation to the unsampled items ($v \notin \hat{\mathbb{V}}^-$), we calculate the auxiliary loss without gradients and refine the loss of item $v_i$ with weight

$$w_i = \frac{-[\hat{P}(v_i|u) \log(\hat{P}(v_i|u)) + (1 - y_i) \log(1 - \hat{P}(v_i|u))]}{\sum_{v_i \in \mathbb{V}} -[\hat{P}(v_i|u) \log(\hat{P}(v_i|u)) + (1 - y_i) \log(1 - \hat{P}(v_i|u))]}. \tag{16}$$

Then we can refine the loss function as:

$$\hat{\mathcal{L}} = \sum_{v_i \in \mathbb{V}^+ \cup \hat{\mathbb{V}}^-} -w_i[y_i \log(\hat{y}_i) + (1 - y_i) \log(1 - \hat{y}_i)]. \tag{17}$$

Items with greater loss over the entire item pool will be taken into account more. Sampled items that used to have higher softmax probabilities have also been corrected.

## 5 EXPERIMENTS

In this section, we conduct two offline experiments on real-world datasets and compare CDDS with existing methods. In addition, online experiments by deploying our method in an industrial advertising system demonstrate effectiveness and efficiency.

### 5.1 DATASETS DESCRIPTIONS

We choose a public representative dataset ML-25M (Harper & Konstan, 2015) and a real-world industrial advertising dataset for evaluating retrieval performance.

- ML-25M[2]. This dataset describes 5-star rating and free-text tagging activity from a movie recommendation service.

---

[2]https://grouplens.org/datasets/movielens/

Table 1: Statistics of datasets

| Statistics | ML-25M | Industrial advertising dataset |
|---|---|---|
| # of training examples | 3,215,856 | 32,189,152 |
| # of test examples | 355,406 | 332,344 |
| # of items | 62,423 | 1,249,013 |

- Industrial advertising dataset. This dataset contains click, downloading, activation, and purchase behaviors in an online advertising system that serves more than a billion users.

**Preprocessing on ML-25M.** Given a user, its ratings on items in chronological order form the session sequence. We split each session to generate feature sequences and labels. For example, for a session $[v_1^{(s)}, v_2^{(s)}, ..., v_n^{(s)}]$, we generate a series of sequences and feedbacks $([v_1^{(s)}], v_2^{(s)})$, $([v_1^{(s)}, v_2^{(s)}], v_3^{(s)})$, $([v_1^{(s)}, v_2^{(s)}, ..., v_{n-1}^{(s)}], v_n^{(s)})$, where $[v_1^{(s)}, v_2^{(s)}, ...v_{n-1}^{(s)}]$ is the generated input sequence and $v_n^{(s)}$ is the feedback item. 5-star rating items are treated as positive feedback items and training positives. We construct the examples with only training positives while training negatives are sampled in the training phase. The movie tags applied by the given user before $v_n^{(s)}$ rated form the tag sequence feature. User id feature, rating sequence feature, and tag sequence feature form user features. Item id feature, genres feature, genome scores, and genome tags form item features. We use all $62,423$ items in the training and testing, and filter out the training examples that have neither rating sequence nor tag sequence. Samples with timestamps before 1514500000 are training samples and samples after 1514500000 are test samples, as the test samples account for about 10 percent of the total samples.

**Preprocessing on industrial advertising dataset.** We classify user feedback into three categories: exposure, click, and conversion. Conversion is a kind of feedback with less ambiguity, such as download, registration, and purchase. The user's most recent conversion is collected as the training positive. Same with the preprocessing on ML-25M, only positive samples are included. Our advertising dataset has richer and more sparse features. We select the examples of a certain day as the training set and sample one percent of the examples from the next day as the test set.

The statistics of the datasets are shown in Table 1.

## 5.2 EVALUATION METRICS

We use different metrics for the offline and online phase. For the offline evaluation, we apply the widely used metrics hit ratio (HR) and mean reciprocal rank (MRR) to measure the performance of the retrieval. In the online phase, the metrics are listed as follows:

- Gross Merchandise Volume (GMV). GMV is the total amount of sales on the advertisers' platform. This metric measures the real income of advertisers.

- Adjust Cost. The cost is the money that advertisers need to pay for advertising platform. The Adjust Cost can be regarded as the real income brought by the experimental strategy after adjustment based on the actual cost.

## 5.3 COMPARISON METHODS

To demonstrate the performance of our proposed method, we use the same two-tower architecture to encode all queries and items. We compared our method with other baseline methods as well as recent state-of-the-art methods:

- **MLP with Sofxmax.** MLP with sofxmax uses a extreme multi-class classification architecture and outputs item probability with a two-layer DNN. This approach only uses the id feature.

- **Batch negatives sampling.** Batch negatives sampling uses the positive items in a mini-batch as shared negatives for all queries in the same batch.

- **Uniform sampling.** Uniform sampling uniformly samples negatives.

Table 2: The performance comparison over ML-25M dataset. Best scores are highlighted in bold-face.

|  | HR@1 (MRR@1) | HR@10 | HR@100 | MRR@10 | MRR@100 |
|---|---|---|---|---|---|
| MLP with softmax | 0.0257 | 0.1072 | 0.3420 | 0.0462 | 0.0538 |
| Batch negatives sampling | 0.0243 | 0.0974 | 0.3006 | 0.0428 | 0.0495 |
| Uniform sampling | 0.0144 | 0.1197 | 0.4588 | 0.0395 | 0.0507 |
| MNS | 0.0311 | 0.1369 | 0.4474 | 0.0572 | 0.0675 |
| ANCE | 0.0309 | 0.1404 | 0.4509 | 0.0582 | **0.0685** |
| CDDS w/o $\hat{\mathcal{L}}$ | 0.0302 | 0.1364 | 0.4462 | 0.0567 | 0.0669 |
| CDDS | **0.0324** | **0.1425** | **0.4637** | **0.0603** | **0.0685** |

- **MNS.** Yang et al. (2020) improve the in-batch sampling by additionally sampling global uniform negatives.
- **ANCE.** Xiong et al. (2020) construct global negatives from the entire item pool by using an asynchronously updated ANN index.
- **CDDS w/o $\hat{\mathcal{L}}$.** The comparison method introduce uniform negatives, batch negatives, and high divergence negatives. Compared with CDDS, the loss function is $\mathcal{L}$ which doesn't alleviate the training-inference inconsistency.

## 5.4 PARAMETER CONFIGURATION

The embedding dimension is set to 64 and the size of mini-batch is set to 512. We use Adam Optimizer (Kingma & Ba, 2014) with the learning rate 0.001. The number of uniform negatives is set to 512 on ML-25M and is set to 16384 on the industrial advertising dataset for Uniform sampling, MNS, and CDDS. With the relative large-scale of uniform negatives, most items are trained adequately on ML-25M. Due to the limitation of GPU allocation, we cannot sample more negative samples to achieve the same effect on the industrial advertising dataset. The number of high divergence negatives is set to 64 for ANCE and CDDS. We use Transformer (Vaswani et al., 2017) to model the sequence feature.

## 5.5 PERFORMANCE COMPARISON

Table 2 illustrates the overall performance on ML-25M dataset, with the best results highlighted in boldface. Table 3 illustrates the performance on the industrial advertising dataset and only HR@K is shown while the online service applies a pre-ranking stage to sort the candidates from different retrieval models. Compared to other methods, our method achieves the best performance.

From table 2 and 3, we observe that batch negatives sampling performs poorly as a result of the vast majority of long-tail items have not been adequately learned. Uniform sampling with rich features achieves the best HR@100 among the comparison methods on ML-25M. The above phenomenon shows that uniform sampling works well on an item pool with $64,263$ movies. As all items are trained adequately, the uniform sampling successfully solves the training-inference inconsistency and performs well in small-scale retrieval. Since popular items and high divergence negatives are not specifically trained, the MRR metrics of the uniform sampling method are not good enough. With a million-scale item pool, the uniform sampling method does not achieve a good performance on the industrial advertising dataset as the items can not be trained adequately. MNS performs better than all of the existing methods at HR@1, which indicates the importance of mixture negatives. ANCE learns high gradients negatives from the entire item pool and achieves a good performance at MRR@10 and MRR@100. This means that learning with high divergence negatives helps to distinguish the rank order in the retrieval candidates. CDDS without $\hat{\mathcal{L}}$ shows that the simple mixture of negatives is not enough and alleviating the training-inference inconsistency is important. From table 3, we observe that CDDS achieves a fast learning convergence while the uniform sampling has not converged over 12 hours to solve the training-inference inconsistency.

From Figure 2, we can observe that the popular items are minority and the long-tail items are major-ity with a Gini index 0.97 on the ML-25M training dataset. The hot items are with higher HR@100 as the hit number at top 100 is closer to the number of testing samples while the long-tail items are

Table 3: The performance comparison over industrial advertising dataset. Best scores are highlighted in boldface.

| | HR@1 | HR@100 | HR@1000 | HR@2000 | Convergence time |
|---|---|---|---|---|---|
| MLP with softmax | - | - | - | - | - |
| Batch negatives sampling | 0.0314 | 0.1683 | 0.3531 | 0.4147 | 4.3h |
| Uniform sampling | 0.0321 | 0.1916 | 0.3946 | 0.4613 | over 12h |
| MNS | 0.0492 | 0.2394 | 0.4496 | 0.5162 | 6.0h |
| ANCE | 0.0489 | 0.2360 | 0.4561 | 0.5224 | 6.6h |
| CDDS w/o $\hat{\mathcal{L}}$ | 0.0476 | 0.2385 | 0.4531 | 0.5225 | 7.4h |
| CDDS | **0.0528** | **0.2443** | **0.4629** | **0.5269** | 5.6h |

with lower HR@100. CDDS has a high recall accuracy for popular items and a large hit number for long-tail items, which shows that our method does not purely retrieve popular items.

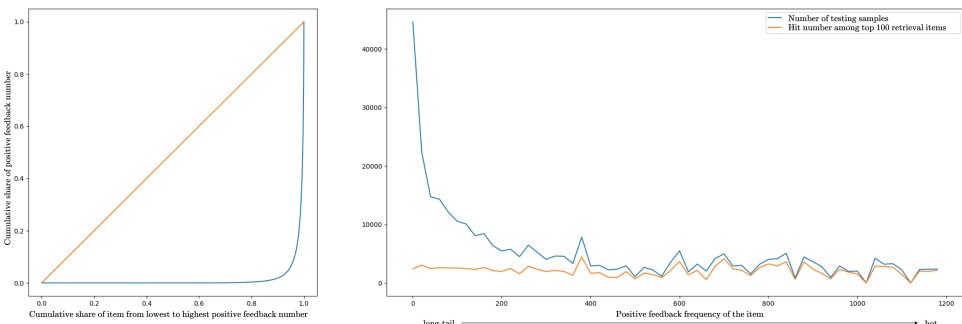

Figure 2: The left part is the Lorenz curve which represents the cumulative proportion of ordered items from lowest to highest in terms of the cumulative proportion of their positive feedback number on ML-25M training dataset. The right part is the CDDS performamce of long-tail items and hot items on the ML-25M test dataset.

### 5.6 ONLINE EXPERIMENTS

We conduct online experiments by deploying our method to handle real traffic in an industrial advertising system. Our experiments start with 1% randomly selected users and finally take effect across the entire traffic. There are two thousand candidate items retrieved by our method. GMV and adjust cost are used to measure the performance of serving online traffic. With the control group consisting of 50% randomly selected users, the experiment in a scenario achieved 0.30% improvement on GMV and 0.14% improvement on adjust cost, and another scenario achieved 0.47% improvement on GMV and 0.27% improvement on adjust cost.

### 6 CONCLUSION

In this work, we propose a novel negative sampling strategy Consistent Data Distribution Sampling to solve the training-inference inconsistency of data distribution brought from sampling negatives. Specifically, we employ a relative large-scale of uniform training negatives and batch negatives to adequately train long-tail and hot items respectively, and employ high divergence negatives to improve the learning convergence. To make the above training samples approximate the serving item data distribution, we introduce an auxiliary loss based on an asynchronous item embedding matrix over the entire item pool. Offline and online experiments show the effectiveness and efficiency of CDDS.

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
