# OpenReview forum: "Consistent Data Distribution Sampling for Large-scale Retrieval"
_ICLR.cc/2023/Conference — Submitted to ICLR 2023_

### Official Review · Reviewer_WYde · 2022-10-23

**Confidence:** 3
**Correctness:** 2
**Technical Novelty And Significance:** 2
**Empirical Novelty And Significance:** Not applicable
**Recommendation:** 3

**Clarity, Quality, Novelty And Reproducibility:**

I had a hard time understanding the paper. I think it needs to have a large overhaul before it can be accepted to any venue/journal.


**Strength And Weaknesses:**

Strengths:
The paper shows a positive effect of the solution in the experiments.
Weaknesses:
1) Paper quality is has big issues. It needs to be proofread, the plots are hard to understand, terminology is not clear.  I think it is enough for rejection by itself.
2) I am not sure if the uniform weight sampling is actually a desired outcome. E.g. in simclr the quality of the results dropped with the batch size after certain batch size.
3) it is not clear what is the baseline in the a/b test.

**Summary Of The Paper:**

The paper proposes a method for consistent distribution sampling in latent two-tower models for recommender system. The method adds weights sampling in the loss calculation (on the mix of subsampled random negatives and negatives from a hard negative mining) to better approximate the overall sampling distribution which seem to help on few open-sourced datasets and in a/b tests in production.

**Summary Of The Review:**

I think the paper has big problems with presentation, so it should be rejected on this basis. I also feel like this paper would be a better fit for a recsys conference (or a workshop), rather than ICLR.

---

### Official Review · Reviewer_RorQ · 2022-10-30

**Confidence:** 4
**Correctness:** 3
**Technical Novelty And Significance:** 2
**Empirical Novelty And Significance:** 2
**Recommendation:** 5

**Clarity, Quality, Novelty And Reproducibility:**

-The paper is mainly written well and easy to follow.

-I am not impressed by either the novelty of the technique or the results. It seems like a fairly incremental paper studying a critical problem.

**Strength And Weaknesses:**

Strengths:

-It provides a good overview of the problem analytically and why we need to be mindful in sampling negatives. It also provides a rigorous analysis of the convergence.

-The results are decent. Particularly close to the ANCE. It is not surprising since it also provides a similarity/distance-dependent process.

Weakness:

-It is not clear the complication and analysis of what similarity check brings concerning complexity (as an incremental unit) and how it grows with the dimensionality of data and representation size. For example, how much does it worsen the training time? What is the trade-off with respect to increased training time (so in an online system, maybe you will be able to retrain 4 times instead of 10 times a day).

**Summary Of The Paper:**

Online ML systems usually have explicit positive samples (E.g., ads conversion, click, buy, like, etc.) to train on. However, negative samples are often implicit and have a much higher volume concerning positives. A common technique in constructing a training data set is to sample from the negatives to having more reasonably balanced data sets. However, the sampling process introduces problems concerning selection bias, not considering the long tail, etc. The paper presents a new negative sampling strategy called consistent data distribution sampling to improve this process. It focuses on the long tail and hot items specifically and provides mechanisms to improve convergence.


**Summary Of The Review:**

I am borderline with this paper. The novelty and results are not impressive. It is an important problem and has strength in analytical analysis. But in such a practical problem, we either need a more thorough practical evaluation or a stronger algorithm with stronger results to pass the bar. so my rating is a marginally below acceptance.

---

### Official Review · Reviewer_sTd5 · 2022-10-31

**Confidence:** 4
**Correctness:** 3
**Technical Novelty And Significance:** 2
**Empirical Novelty And Significance:** 2
**Recommendation:** 3

**Clarity, Quality, Novelty And Reproducibility:**

The paper is well organized and I believe the experiments can be reproduced. However, the quality and novelty are incremental. To highlight the original contribution, authors should figure out the difference between this submission and the previous work, like Xiong et al. (2020), Yang et al. (2020).

**Strength And Weaknesses:**

Strength:
1. The topic is interesting and important to improve the user’s experiences in industrial system.
2. The submission has an online experiment to validate the effectiveness of the model in real industrial system.

Weakness:

1. The novelty and contribution are incremental. The main contribution is to propose the loss function based on the classical binary-cross entropy by weighting each training items.
2. The experiments are not adequate. Please conduct more experiments on other datasets and make up some ablation studies.
3. Please give more explanations for the auxiliary loss (i.e. e.q. (10) and e.q. (15),). What’s the intuitive meaning of the auxiliary loss?  How about the training efficiency as the auxiliary loss and the weight (i.e. e.q.(16)) consider all the items?
4. In the last paragraph on page 5, the sentence ‘Along with the training, the difference between $\theta$ and $\theta^{‘}$ becomes larger. Intuitively, the difference between $\theta$ and $\theta^{‘}$ should be smaller as the model converges. Please elaborate on how to set the checkpoints and how to update the checkpoint in your experiments.


**Summary Of The Paper:**

Long-tail distribution of items is an important factor which affects the user’s experience in industrial search systems, recommendation systems, and so on. This submission tries to address the training-inference inconsistency due to the long-tail distribution. Concretely, the authors comprise a relatively large-scale of uniform negative sampling and batch negative sampling to make sure that long-tail items and hot items can be trained adequately. To faster the convergence of training, the proposed model also searches some hard negative items based the ANN index for the training queries. The main contribution of this submission is to propose a new loss function which considers the weight of each positive and negative training items. Both the offline and online experiments validate the effectiveness of the proposed model.

**Summary Of The Review:**

The submission focuses on the training of long-tail distribution of items which is important to improve the user’s experience in real search systems and recommendation systems. However, the novelty and the original contribution are incremental. Additionally, the experiments are not adequate.

---

### Official Review · Reviewer_ExGn · 2022-11-01

**Confidence:** 3
**Correctness:** 3
**Technical Novelty And Significance:** 2
**Empirical Novelty And Significance:** 2
**Recommendation:** 5

**Clarity, Quality, Novelty And Reproducibility:**

*clarity*
- How many steps that the ANN index table is behind the training? It would need enough time to allow the inference for index.
If the step gap is large. This may yield suboptimal performance since the embedding may change a lot from the lookup table. The step gap matters. it would be nice to have one experiment around this discussion
- Wonder to see the ablation of sampled negatives, like CDDS vs CDDS - high divergence negative vs CDDS - uniform negative vs etc

*writting*

- Figure 1 is never cross-referred in the main text, it would be nice to describe in detail in section 4.
- Section 5.6 seems not necessary as it does not compare with any other approaches.
Some of my guesses of what authors want to express,
- procession -> processing, procession mostly means parade
- Totally - > In general
- ”As a result of retrieving a possible...., the training negative set ........“ I don't see a causation between two sub parts, please simply say V^- = V \ V^+. That is good enough

**Strength And Weaknesses:**

*Strength*
Offline experiments compared with related approaches and peer work show SOTA
Combining multiple sampling strategy and the auxiliary loss is somewhat novel

*Weaknesses*
- The experimental results are thorough, but lack of some ablations, see clarity section, and it is also necessary to support the claim that "high divergence improves the learning convergence".
- The results seems marginal, authors can clarify what is the typical gain in published papers for such datasets as an auxiliary support.
- The difference and novelty of the proposed approach is not highlighted, and it is confusing that what makes a difference. Please include concrete technical contribution in the introduction section, current presentation is a bit high level. e.g. "We proposes CDDS which .... by tech 1, tech 2, tech 3......"
- The writing of the paper needs improvement. Especially the incorrect clause usage/unnecessary inverted order sentence improve the difficulty of reading. Simple and short SVO sentences are fine. And please use accurate words to express.


**Summary Of The Paper:**

This work tries to tackle negative sampling in embedding model training for retrieval. Consistent Data Distribution Sampling is proposed by combining large-scale uniform training negatives with batch negatives. High divergence negatives are employed to improve the learning convergence. The sampled negatives are fused in a proposed auxiliary loss based on the asynchronized embedding matrix over the entire index set. The experiments show SOTA performance.

**Summary Of The Review:**

The technical part of the paper is somewhat sound, but the presentation of the work needs improvement (not only grammatically). The skeleton of the work lies on prior works, unless authors clarify the difference from prior works, I think the novelty is limited.

---

### Decision · Program_Chairs · 2023-01-20

**Decision:**

Reject

**Justification For Why Not Higher Score:**

No rebuttal by the authors.

**Justification For Why Not Lower Score:**

N/A

**Metareview: Summary, Strengths And Weaknesses:**

The paper proposes Consistent Data Distribution Sampling (CDDS) for mitigating the bias merging from negative sampling. The paper does show SOTA results. The paper does not places itslef in the context of existing prior work and how their approach is distinctive and differs from other approaches.